# Comparative Studies on the Antioxidant Profiles of Curcumin and Bisdemethoxycurcumin in Erythrocytes and Broiler Chickens

**DOI:** 10.3390/ani9110953

**Published:** 2019-11-11

**Authors:** Jingfei Zhang, Hongli Han, Mingming Shen, Lili Zhang, Tian Wang

**Affiliations:** 1College of Animal Science and Technology, Nanjing Agricultural University, No. 6, Tongwei Road, Xuanwu District, Nanjing 210095, China; zhangjingfei@njau.edu.cn (J.Z.); 2018105060@njau.edu.cn (H.H.); 2017105056@njau.edu.cn (M.S.); zhanglili@njau.edu.cn (L.Z.); 2National Experimental Teaching Demonstration Center of Animal Science, Nanjing Agricultural University, No. 6, Tongwei Road, Xuanwu District, Nanjing 210095, China

**Keywords:** curcumin, bisdemethoxycurcumin, erythrocytes, broiler, redox potential

## Abstract

**Simple Summary:**

Turmeric, which is the rhizome of *Curcuma longa*, has a long history for spice and medicine in China, India, and other tropical countries. Curcuminoids, as the principle active compounds of turmeric, consist of curcumin (about 77%), demethoxycurcumin (about 17%), and bisdemethoxycurcumin (about 3%). Studies showed that curcuminoids, especially curcumin, possesses antioxidant, free radical scavenging activities, and thus have a health-promoting effect in human and animals. Of the three curcuminoids, extensive research on the biological activity of curcumin was carried out for decades. However, its natural analogues bisdemethoxycurcumin was relatively less investigated. Based the records, there was still controversy regarding the relative potency of antioxidant activity of curcuminoid that is dependent on different cell types and animal models, which ultimately affected their beneficial effects on the intestinal health and animal production as well. Thus, whether curcumin and bisdemethoxycurcumin shared the same efficiency of antioxidant activity in chicken erythrocytes and broiler chickens remains unknown. Our results demonstrated, for the first time, that the bisdemethoxycurcumin, acting like curcumin, exerted good free radical scavenging activity in erythrocytes and improved the redox status in broilers, although there were some slight differences in their efficiency of antioxidant activities in broiler chickens.

**Abstract:**

The aim of this study was to investigate the antioxidant effects of curcumin and bisdemethoxycurcumin in both 2,20-azobis(2-amidinopropane) dihydrochloride (AAPH)-treated erythrocytes and broiler chickens. In experiment 1, chicken erythrocytes were employed to determine the antioxidant protection against AAPH treatment. Significant differences in hemolysis, superoxide dismutase (SOD) activity, and malondialdehyde (MDA) content were observed between the control and curcuminoids-treated groups. In experiment 2, a total of 480 Arbor Acres broilers with the similar body weights were used. All of the birds were fed basal diet and basal diet with 150 mg/kg curcumin or bisdemethoxycurcumin, respectively. The results showed that curcuminoids significantly increased ADG, serum antioxidant capacity, the glutathione redox potential of small intestine, the gene expression of Nrf2, and its related antioxidant enzymes. Besides, curcumin and bisdemethoxycurcumin increased the antioxidant activities of serum, diet, and excreta while using the 2,2′-azinobis(3-ethylbenzothiazoline-6-sulfonic acid) diammonium salt and ferric-reducing antioxidant power methods. It was concluded that bisdemethoxycurcumin, acting like curcumin, exerted good free radical scavenging activity in erythrocytes and improved the redox status in broilers, although there were some slight differences in their efficiency of antioxidant activities.

## 1. Introduction

Turmeric, which is the rhizome of *Curcuma longa*, has a long history for spice and medicine in China, India, and other tropical countries. More than sixty active compounds have been isolated and identified from turmeric [1,2,3]. Studies showed curcumin, accounting for up to 77%, as the most important active compound of turmeric, demethoxycurcumin took second place up to 17%, and bisdemethoxycurcumin had the least about 3% [4,5]. Research demonstrated that curcumin, demethoxycurcumin and bisdemethoxycurcumin share the same β-diketone bridge but differed the number and position of methoxy groups on two aromatic rings [6,7]. They showed great prospect for application in the medicine, food, and animal industry due to their high safety and wide spectrum of biological activities. Collectively, all three of them were referred to curcuminoids, which were also the major components of commercial curcumin product. 

Of the three curcuminoids, extensive research on the biological activity of curcumin was carried out in animal and cell models for decades. However, little attention was focused on the beneficial effects of demethoxycurcumin and bisdemethoxycurcumin. Ahsan et al. showed that demethoxycurcumin and bisdemethoxycurcumin had antioxidant activity that was similar to that of curcumin, which protected DNA against oxidant damage [8]. Recent studies reported that, although all three are highly reactive in neutralization of free radicals and induction of antioxidant system, there were significant differences in their antioxidant activities. In lipid peroxidation, the efficacy of demethoxycurcumin and bisdemethoxycurcumin in scavenging free radicals were much stronger than curcumin. In the chemical system, the free radical scavenging activity significantly decreased in the order: curcumin > demethoxycurcumin > bisdemethoxycurcumin [9]. In the cell model of microglia, curcuminoid played similar protective roles against LPS-induced pro-inflammatory reaction, although cells with bisdemethoxycurcumin treatment were more sensitive to the oxidant injury when compared to that of curcumin [10]. It was suggested that there was still controversy regarding the relative potency of antioxidant activity of curcuminoid dependent on different cell types and animal models, which ultimately affected their beneficial effects on the intestinal health and animal production. 

Although the antioxidant effects of curcumin have been widely reported in vitro and in vivo models, its natural analogues bisdemethoxycurcumin was relatively less investigated. Whether curcumin and bisdemethoxycurcumin shared the same efficiency of antioxidant activity in chicken erythrocytes and broiler chickens remains unknown. In addition, the antioxidant mechanism of curcuminoids in the regulation of redox balance, and the antioxidant activity on diet, excreta, serum, and small intestinal contents while using the 2,2′-azinobis(3-ethylbenzothiazoline-6-sulfonic acid) diammonium salt (ABTS) and ferric-reducing antioxidant power (FRAP) methods were largely unexplored. 

Therefore, the aims of this study were, first, to investigate the comparative effects of curcumin and bisdemethoxycurcumin on the antioxidant protection in 2,20-azobis(2-amidinopropane) dihydrochloride (AAPH)-treated erythrocytes (Experiment 1). Subsequently, we investigated the effect of curcumin and bisdemethoxycurcumin on growth performance, serum antioxidant capacity, glutathione redox potential, and the expression of NF-E2-related factor 2 (Nrf2) and other antioxidant genes in broiler chickens (Experiment 2). It was hypothesized that curcumin and bisdemethoxycurcumin protected erythrocytes and increased the free radicals scavenging activities in different tissues. We further hypothesized that the supplementation of curcumin and bisdemethoxycurcumin decreased the redox potential, potentiate the Nrf2 gene expression, and the maintained the redox balance in vivo. 

## 2. Materials and Methods

### 2.1. Chemicals 

Curcumin and bisdemethoxycurcumin were provided by KeHu Biotechnology Research Center (Guangzhou, People’s Republic of China), whose purity was 97 to 98%, as determined by high performance liquid chromatography analysis [11]. AAPH were purchased from Sigma Chemical Co., Ltd. All of the other chemicals used in the present study were of analytical grade and obtained from Shanghai Chemical Agents Company, China. 

### 2.2. Chicken Erythrocytes Preparation 

In experiment 1, the whole blood units from a single male broiler line were freshly collected in heparin-containing tubes. The plasma and buffy coat were removed together with the supernatant by centrifuging at 1500× *g* for 10 min. The erythrocytes were washed three times with the cool PBS buffer (pH = 7.4, containing 150 mM NaCl, 1.9 mM Na_2_HPO_4_, and 8.1 mM NaH_2_PO_4_). Subsequently, the same PBS buffer prepared 2% suspension of chicken erythrocytes for further analysis. 

AAPH was employed as an aqueous peroxyl radical generator in the present study to evaluate and compare the antioxidant effects of curcumin and bisdemethoxycurcumin to protect erythrocytes against oxidant injury. Briefly, 2% erythrocytes suspension was pre-incubated for 30 min. at ambient temperature with curcumin or bisdemethoxycurcumin in different concentrations at 0.5 μM, 1 μM, 5 μM, 10 μM, and 20 μM, and followed by incubation with or without AAPH (final concentration 75 mM) for 5 h under gentle shaking. 

### 2.3. Determination of Hemolysis 

The rate of hemolysis was spectrophotometrically determined according to the method of Zhang et al. [12]. After 5 h of incubation, two aliquots of the treated erythrocytes were taken out and centrifuged at 1500× *g* for 10 min. to obtain red blood cell pellets. The erythrocytes suspension was prepared by saline (1:7, v/v, 150 mM). The percentage of hemolysis was determined by measuring the absorbance of the supernatant (A) at 540 nm, relative to that of the complete hemolysis (B) induced by distilled water. The percentage of hemolysis was calculated while using the fowling formula: Hemolysis (%) = A/B * 100. Three replications of each treatment were used for these calculations. 

### 2.4. Determination of Superoxide Dismutase Activity and Malondialdehyde Levels in Erythrocytes

After the exposure, erythrocytes were broken by repeated sonication for 15 s each in an ice-water bath. The superoxide dismutase (SOD) activity, malondialdehyde (MDA) levels, and hemoglobin contents in erythrocytes were determined with commercial kits (Nanjing Jiancheng Insititute of Bioengineering, Nanjing, China), according to the instructions of the manufacturer. The result of SOD activity was expressed as units (U) per gram of hemoglobin (U/g Hb) and MDA level was expressed as the micromole per gram of hemoglobin (μmol/g Hb) in erythrocytes.

### 2.5. Animals, Diets, and Experimental Design

The Ethical Committee approved this experimental protocol and it was conducted under the supervision of the Institutional Animal Care and Use Committee of Nanjing Agricultural University, Nanjing, China.

In experiment 2, a total of 480 Arbor Acres broilers with the initial body weights ranging from 46.2 g to 48.4 g were purchased from a commercial hatchery (Hefei, Anhui, People’s Republic of China) and used in this experiment. All of the birds were randomly allocated to three dietary treatments with eight replicates and 20 birds per replicate. Birds in the three treatment groups obtained the basal diet (CON group), and the basal diet with an additional 150 mg/kg curcumin (CUR group) or bisdemethoxycurcumin (BUR group), respectively. All of the birds were placed in wire cages with a three-level battery and kept in an environmentally controlled room that was maintained at 34−36 °C until the birds reached 14 days of age and then gradually decreased to 26 ± 1 °C by 21 days of age, after which it was maintained at room temperature (24−26 °C) until the end of the experiment. The basal diets were formulated in accordance with the NRC (1994) guidelines for meeting the nutrient requirements of broilers. Table 1 shows the diet compositions.

### 2.6. Sample Collection 

At 42 days of age, one bird was randomly selected from each replicate, electrically stunned, and then slaughtered by cervical dislocation. The blood samples were collected in an anticoagulant-free vacutainer tube. The serum was then obtained by the centrifugation with 3000× *g* for 10 min. and stored at −20 °C for further analysis. The liver, duodenum, jejunum, and ileum tissues were rapidly excised and washed with 0.86% sodium chloride buffer. Small portions of these tissues were then frozen with liquid nitrogen, and stored at −80 °C for further analysis. Excreta samples were obtained on the clean plastic collection trays that were placed under each replicate cage. All of the feed, digesta of small intestine, and excreta samples were dry-ashed and then stored at 4 °C for antioxidant analysis. 

### 2.7. Bird Performance 

The birds were weighted per pen at the day of 1, 21, and 42 age. The feed intake was recorded during the three following periods: 1 to 21 d, 22 to 42 d, and 1 to 42 d of age. The average daily body weight gain (ADG), average daily feed intake (ADFI), and feed conversion ratio (FCR) were calculated for the aforementioned periods. The mortality that was included in the calculation of growth performance was registered during the whole experimental period.

### 2.8. Measurement of Serum Antioxidant Index 

The activities of catalase (CAT), total superoxide dismutase (T-SOD), glutathione peroxidase (GPx), total antioxidant capacity (T-AOC), and MDA concentrations in serum were spectrophotometrically determined with the commercial kits (Nanjing Jiancheng Insititute of Bioengineering, Nanjing, China), according to the manufacturer’s protocol. The results of CAT, T-SOD, GPx, and T-AOC activities were expressed as units (U) per milliliter. The MDA concentration was expressed as nanomoles per milliliter. The concentration of the total thiol group (T-SH) was determined according to the previous study and expressed as micromolers per liter [13]. The serum protein carbonyl (PC) levels were determined while using 2,4-dinitrophenylhydrazine according to the method of Reiter et al. [14]. Briefly, the serum was prepared as free from nucleic acids by overnight incubation with streptomycin sulfate. Afterwards, 2,4-dinitrophenylhydrazine and HCl were added to produce the protein hydrazine in the presence of guanidine hydrochloride. The absorbance was recorded at 370 nm and calculated while using the molar extinction coefficient of 22,000. The result of the PC concentration was corrected for the protein concentrations in each sample and expressed as nanomoles per milligram of protein.

### 2.9. Glutathione Redox Potential of Serum, Liver, Duodenum, Jejunum and Ileum 

The samples of the liver, duodenum, jejunum, and ileum were minced and homogenized with an ice-cold buffer containing 0.86% sodium chloride buffer (w/v, 1:9). The supernatant was collected by the centrifugation with 3500× *g* for 10 min. at 4 °C and used in the further analysis. The concentrations of reduced glutathione (r-GSH) and oxidized glutathione (o-GSH) were spectrophotometrically determined with the commercial kits (Nanjing Jiancheng Insititute of Bioengineering, Nanjing, China). The glutathione redox potential (E_h_) values of the serum and tissue were calculated while using the appropriate forms of the Nernst equation (mV) for the respective r-GSH/ o-GSH and pools: E_h_ = −64+30 log ([o-GSH]/[r-GSH]^2^) for pH 7.4 in tissues and serum [15]. 

### 2.10. Real-Time Polymerase Chain Reaction (PCR) Analysis 

Total RNA was isolated from frozen liver sample and small intestine tissues while using Trizol reagent (TaKaRa, Dalian, China) according to the manufacturer’s instructions. The quantity of the total RNA was measured with a Nanodrop ND-2000c spectrophotometer (Thermo Fisher Scientific, Camden, NJ). The cNDA was immediately synthesized by reverse transcription using a PrimeScript RT Reagent kit (Takara Biotechnology Co., Dalian, China).

The expression of genes in liver, jejunum, and ileum samples was measured while using real-time quantitative PCR (RT-qPCR) with SYBR Premix Ex Taq II kit (Tli RNaseH Plus; Takara Biotechnology) and an ABI 7300 Fast Real-Time PCR detection system (Applied Biosystems). RT-qPCR was performed at 95 °C for 30 s, followed by 40 cycles of 95 °C for 5 s and 60 °C for 30 s. All of the samples were run in triplicate, and Table 2 shows the gene-specific primer sequences. The mRNA expression levels of genes were calculated while using the 2^−ΔΔCt^ method and normalized to the value of the reference gene β-actin. The final result of each target gene expression was expressed as the percentage of the CON group. 

### 2.11. Measurement of the ABTS Scavenging Activity and FRAP Value 

The dry-ashed samples of diets, small intestinal content (duodenum, jejunum, and ileum), and excreta were treated with boiling 4 M HCl [16,17]. The samples were extracted by shaking with methanol-water (50:50 vol/vol, 50 mL/g of sample) for 1 h and with acetone-water (70:30 vol/vol, 50 mL/g of sample) for 1 h at room temperature. The supernatants were obtained by the centrifugation with 3000× *g* for 15 min. The ABTS scavenging activity and FRAP value of serum, and the above supernatants were measured according to the methods of Zhang et al. [12].

### 2.12. Statistical Analysis 

Data (experiments 1 and 2) were expressed as the mean ± standard error (SEM). All the data were analyzed by using one-way analysis of variation (ANOVA) by SPSS (SPSS 17.0 for Windows, SPSS, Inc., Chicago, IL, USA). The significance of comparisons was conducted while using the Tukey’s multiple range tests and a *p* value < 0.05 was considered to be statistically significant. 

## 3. Results

### 3.1. Erythrocytes Hemolysis (Experiment 1) 

Figure 1 shows the effects of curcumin and bisdemethoxycurcumin on the hemolysis in AAPH-treated chicken erythrocytes. The incubation of chicken erythrocytes with AAPH caused a marked hemolysis (*p* < 0.05) as compared to the CON group. The hemolysis was decreased in dose-dependent manner, followed by the curcumin and bisdemethoxycurcumin supplementation. The hemolysis were lower (*p* < 0.05) in the CUR and BUR group at the concentrations ranging from 0.5 to 20 μM when compared to the AAPH group.

### 3.2. SOD Activity and MDA Level in Erythrocytes (Experiment 1) 

Figure 2 shows the effects of curcumin and bisdemethoxycurcumin on the SOD activity and MDA level in AAPH-treated chicken erythrocytes. AAPH challenge inhibited (*p* < 0.05) the SOD activities and increased (*p* < 0.05) the MDA levels in chicken erythrocytes. When compared to the AAPH group, the SOD activities were higher (*p* < 0.05), while the MDA levels were lower (*p* < 0.05) in both the CUR and BUR group at the concentrations ranging from 0.5 to 20 μM. Moreover, the MDA levels in the CUR and BUR group showed a declining trend with the increasing concentrations of samples. 

### 3.3. Growth Performance (Experiment 2) 

Table 3 shows the effects of curcumin and bisdemethoxycurcumin on the growth performance in broilers. Supplementation of curcumin increased ADG (*p* < 0.05) as compared to the control group from days 21 to 42 and days 1 to 42. When compared to the CON group, the ADG did not significantly differ from days 21 to 42 and days 1 to 42 in the BUR group (*p* > 0.05). There were no significant differences (*p* > 0.05) on the ADFI and FCR among the three treatment groups. 

### 3.4. Serum Antioxidant Index (Experiment 2) 

Table 4 shows the effects of curcumin and bisdemethoxycurcumin on the serum antioxidant index in broilers. When compared to the CON group, dietary BUN supplementation increased (*p* < 0.05) the serum T-AOC activity. Serum MDA concentrations in the CUR group were lower (*p* < 0.05) as compared to the CON group. There were no significant differences (*p* > 0.05) among the tree treatments regarding the CAT activity, T-SOD activity, GPx activity, T-SH, and PC concentrations in serum.

### 3.5. Glutathione Redox Potential in Different Tissues (Experiment 2) 

Table 5 reports the effects of curcumin and bisdemethoxycurcumin on the glutathione redox potential of serum, liver, and small intestine (duodenum, jejunum, and ileum). When compared to the CON group, the glutathione redox potentials of jejunum were decreased (*p* < 0.05) in both the CUR and BUR group. Supplementation of bisdemethoxycurcumin showed a reduced (*p* < 0.05) glutathione redox potential of ileum in the BUR group as compared to the CON group. The glutathione redox potential of serum, liver, and duodenum did not significantly differ (*p* > 0.05) between the three treatment groups. 

Table 5 also showed the r-GSH and o-GSH concentrations of serum, liver, and small intestine (duodenum, jejunum, and ileum) in broilers. When compared to the CON group, the r-GSH concentrations of liver were higher (*p* < 0.05) than that of the BUR group. Dietary curcumin and bisdemethoxycurcumin supplementation increased (*p* < 0.05) the r-GSH concentrations, while it decreased (*p* < 0.05) the o-GSH concentrations of jejunum as compared to the CON group. There were no different significant differences (*p* > 0.05) on the GSH and GSSG concentrations of serum, duodenum, and ileum between the three treatment groups. 

### 3.6. Antioxidant-Related Gene Expression (Experiment 2) 

Table 6 presents the effects of curcumin and bisdemethoxycurcumin on the antioxidant gene expression of liver, jejunum, and ileum. The ileum *Nrf2* mRNA level was higher (*p* < 0.05) in both the CUR and BUR groups than that of the CON group, whereas the liver Nrf2 mRNA level was higher (*p* < 0.05) only in BUR group. Compared to the CON group, the *HO-1* mRNA level of liver was increased (*p* < 0.05) by bisdemethoxycurcumin supplementation. Dietary curcumin and bisdemethoxycurcumin supplementation increased (*p* < 0.05) the copper and zinc superoxide dismutase (*CuZnSOD*) mRNA levels of ileum as compared to the CON group. 

### 3.7. Antioxidant Activity in Diets, Excreta, Serum, and Small Intestine Contents 

Figure 3 shows the antioxidant activities of diet, excreta, serum, and small intestinal contents (duodenum, jejunum and ileum) using the ABTS and FRAP methods. The serum antioxidant activities, using the ABTS method, were higher (*p* < 0.05) in both the CUR and BUR groups as compared to the CON group. An increase antioxidant activity of diet using the ABTS method was observed in the BUR group as compared to the CON group (*p* < 0.05). The antioxidant activities of excreta and serum, while using the FRAP method, were increased (*p* < 0.05) in both the CUR and BUR groups as compared to the CON group. 

## 4. Discussion 

Redox homeostasis is a stable state that is required for physiological process and cellular metabolism in animal production [18]. When considering the intensive farming of broiler chickens, the burst of free radical oxygen species will be a major threat to the redox homeostasis, thus undermining animal welfare and compromising performance [19,20]. The adaptive response against the redox imbalance is mediated by either endogenous antioxidant system or exogenous compound (i.e., phenolic compounds and natural antioxidants) [21,22]. In the present study, we have demonstrated that curcumin and bisdemethoxycurcumin, two principal curcuminoids found in commercially available turmeric extracts, positively improved the redox system and enhanced the antioxidant capacity of broiler chicken in vitro and *in vivo*. More interestingly, bisdemethoxycurcumin, which displays a similar basic chemical structure with curcumin, differs in its ability to scavenge reactive free radicals in erythrocytes and regulate the redox homeostasis in broilers. On the basis of previous studies and our observation, it was probably due to the differential efficiency of curcuminoids, especially bisdemethoxycurcumin, which indicates that bisdemethoxycurcumin can be a potential candidate as natural antioxidant to promote the animal health [8,9,10]. 

Erythrocytes contain high polyunsaturated fatty acids in membrane and they are vulnerable to free radical exposure due to its role in the oxygen transport [23]. In the present study, our results demonstrated that curcumin and bisdemethoxycurcumin decreased hemolysis, inhibited lipid peroxidation, and restored the SOD activities in AAPH-treated erythrocytes. It is possible that the antioxidant activities of curcumin and bisdemethoxycurcumin contribute most to the protection on erythrocytes. Bisdemethoxycurcumin, acting like curcumin, could directly neutralize free radicals and interrupt the chain reaction [7]. On the other hand, the induction of curcuminoids on the antioxidant enzyme activities, like SOD, has been widely reported [24,25,26]. The increased SOD in AAPH-treated erythrocytes might result from their indirect free radical scavenging property. Although the exact mechanism needs to be further verified, our results have confirmed that bisdemethoxycurcumin is effective in suppressing erythrocyte oxidant damage via its antioxidant activities, which is in agreement with previous reports [7,9]. 

The results showed that curcumin was capable of increasing the serum antioxidant capacity and activating the Nrf2 signaling pathway. The study also showed that a comparable improvement of antioxidant system was produced when birds were fed a bisdemethoxycurcumin-contained diet at the same dose of 150 mg/kg. Curcuminoids, either curcumin or bisdemethoxycurcumin, have been identified as potent Nrf2 activators in clinic and animal models [27,28,29]. When the expression of Nrf2 is induced, serum MDA concentrations and ROS production are decreased by enhancement of phase II detoxifying and antioxidant enzymes [30,31]. Therefore, the antioxidant effects of curcuminoids are due, at least partially, to the stimulation of the Nrf2 signaling pathway. However, unlike curcumin, the potential antioxidant mechanisms of bisdemethoxycurcumin are not fully understood. Kim et al. reported that the induction of bisdemethoxycurcumin on cellular antioxidant defense system represents a crucial instrument of its antioxidant effect, as measured by Nrf2 translocation in rat cerebral ischemia [30]. Pugazhenthi et al. demonstrated that bisdemethoxycurcumin was capable of inducing the expression of HO-1 and other phase II enzymes, including NQO1 [32]. Consistent with the above findings, we provided evidence that curcuminoids, despite structural difference, exerted similar antioxidant property by stimulating the Nrf2 signaling pathway and improving the total antioxidant capacity in chickens.

Although many studies suggested that a compromised performance of animals was connected to an alteration of intracellular redox status, there was little attention on the glutathione redox potential and a causal link between antioxidant administration and the change of redox potential. GSH is the most abundant intracellular thiol antioxidant and it plays important roles in maintaining the redox balance in organism [33]. Studies showed that small changes of reduced and oxidized GSH concentrations would be amplified to large effects on redox potential through the regulation of thiol-disulfide balance on protein [34,35], which made the glutathione redox potential a sensitive redox indicator. Our data revealed the presence of decreased glutathione redox potential upon the increased r-GSH concentrations and decreased o-GSH concentrations. The significant differences of glutathione redox potential were observed in jejunum and ileum, while no changes were observed in the serum, liver, and duodenum following curcumin and bisdemethoxycurcumin administration. These results in animals were consistent with previous observations, indicating that dietary antioxidants administration might be an efficient way to regulate the redox status of cellular microenvironment, including curcuminoids [36,37]. r-GSH accumulation and o-GSH depletion were well-reported events that were detected in curcuminoids-treated animals and cells [38,39]. Whether the shift of redox balance towards reduction is a result of the induction of GSH system or the enhanced total antioxidant capacity by curcuminoids administration needs further study. 

We also determined the antioxidant contribution of curcuminoids in diet, excreta, serum, and small intestinal content. The free radical scavenging activities of curcuminoids were reported in various in vitro assays [7,9]. ABTS radical was dissolved in both aqueous and organic solution, and has been widely used to evaluate radical quenching capacities of compounds [40]. The FRAP assay, which was characterized by high reproducibility and easy operability, was conducted based on a redox-linked colorimetric reaction [41]. Our results showed that both curcumin and bisdemethoxycurcumin improved the serum antioxidant activity using the ABTS and FRAP methods. Bisdemethoxycurcumin increased the antioxidant activity of diet and excreta while using the ABTS and FRAP methods, while curcumin increased the FRAP values of excreta. Holder et al. [42] reported that curcumin followed oral and intraperitoneal administration in rats were primarily excreted in faece within 24 h. Faece shows a higher pH values than intestinal content due to the inclusion of large intestinal secretion and partial intestinal bacteria. Under near to neutral and weak basic conditions, curcumin was unstable and it rapidly degraded to ferulic acid and feruloylmethane [43]. Ferulic acid was potent in many biological activities, including the antioxidant activity and even proven better to curcumin according to the trolox equivalent antioxidant capacity assay, FRAP assay, and oxygen radical absorbance capacity assay [44]. The increased antioxidant capacity of excretion after curcuminoids administration could be partly explained by curcumin and its metabolized products.

## 5. Conclusions

The present study characterized the antioxidant activities of curcuminoids (curcumin and bisdemethoxycurcumin) in both erythrocytes and broiler chickens. Moreover, the treatment of curcumin and bisdemethoxycurcumin increased the antioxidant activity of diet and excreta while using the ABTS and FRAP methods. Interestingly, our results provided evidence that bisdemethoxycurcumin and curcumin had similar antioxidant activities, although there was some difference on the detailed antioxidant index, such as serum MDA concentration, the glutathione redox potential, and the Nrf2 gene expression of ileum, and highlighted the potential role of bisdemethoxycurcumin being used as a dietary antioxidant additive in animal feed.

## Figures and Tables

**Figure 1 animals-09-00953-f001:**
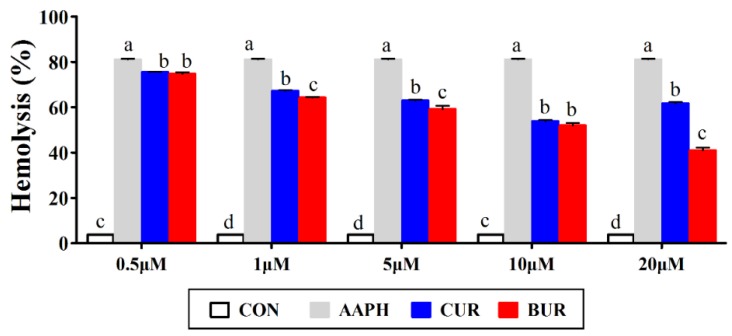
Effects of curcumin and bisdemethoxycurcumin on the hemolysis in 2,20-azobis(2-amidinopropane) dihydrochloride (AAPH)-treated erythrocytes (Experiment 1). Results are expressed as the mean ± SEM. Columns labeled with different letters are significantly different (*p* < 0.05). CON group, the normal chicken erythrocytes; AAPH group, the chicken erythrocytes incubated with AAPH; CUR, the chicken erythrocytes pre-incubated with curcumin and followed by AAPH treatment; BUR group, the chicken erythrocytes pre-incubated with bisdemethoxycurcumin and followed by AAPH treatment.

**Figure 2 animals-09-00953-f002:**
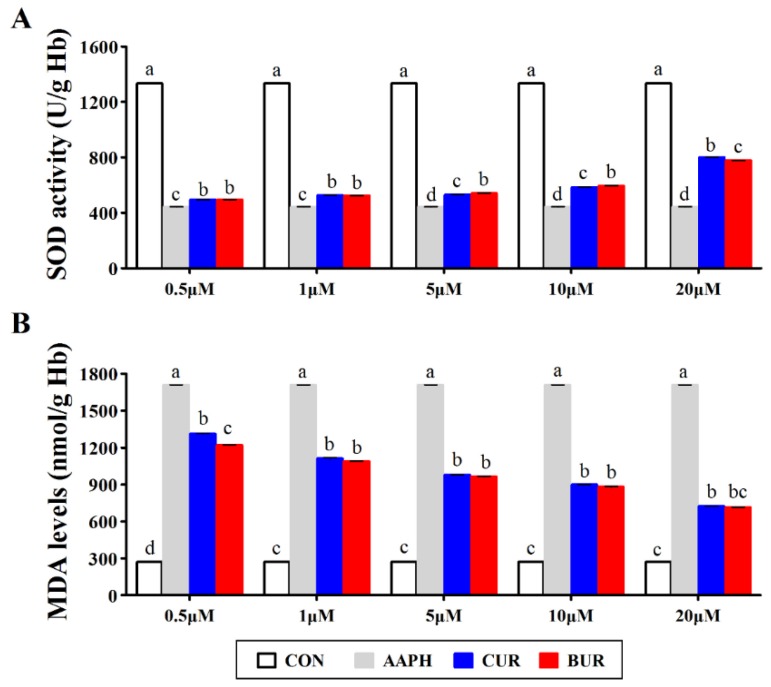
Effects of curcumin and bisdemethoxycurcumin on (**A**) the superoxide dismutase (SOD) activities and (**B**) malondialdehyde (MDA) levels in AAPH-treated erythrocytes (Experiment 1). Results are expressed as the mean ± SEM. Columns labeled with different letters are significantly different (*p* < 0.05). CON group, the normal chicken erythrocytes; AAPH group, the chicken erythrocytes incubated with AAPH; CUR, the chicken erythrocytes pre-incubated with curcumin and followed by AAPH treatment; BUR group, the chicken erythrocytes pre-incubated with bisdemethoxycurcumin and followed by AAPH treatment.

**Figure 3 animals-09-00953-f003:**
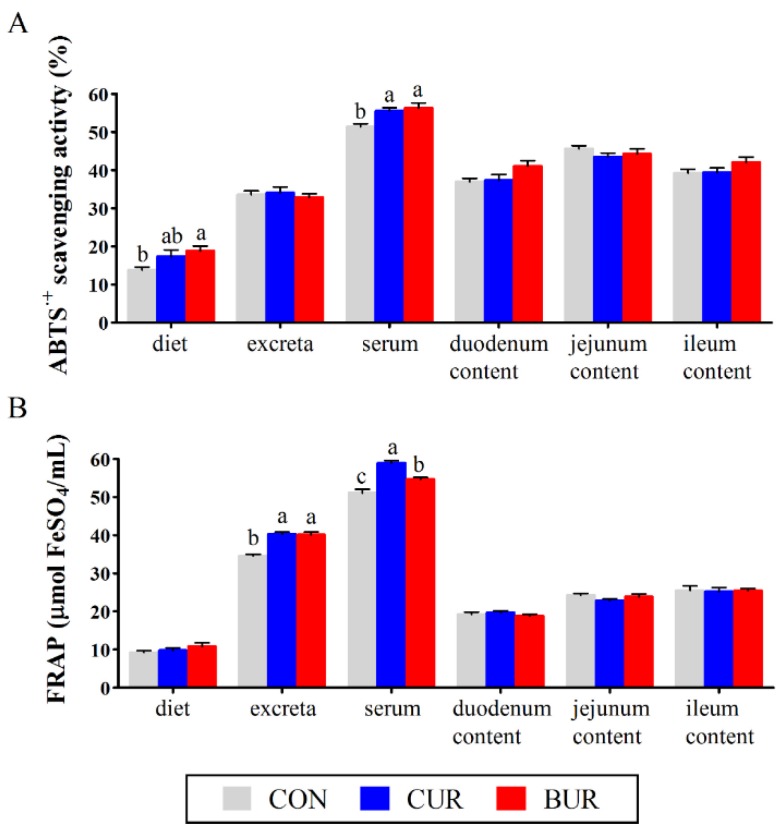
Effects of curcumin and bisdemethoxycurcumin on the antioxidant capacity of diets, excreta, serum, and small intestinal content (duodenum, jejunum and ileum) while using the (**A**) ABTS and (**B**) FRAP methods in broiler chickens (Experiment 2). Results are expressed as the mean ± SEM. Columns labeled with different letters are significantly different (*p* < 0.05). CON group, birds were fed the basal diet; CUR group, birds were fed the basal diet with 150 mg/kg curcumin; BUR group, birds were fed the basal diet with 150 mg/kg bisdemethoxycurcumin.

**Table 1 animals-09-00953-t001:** Ingredient Composition and Calculated Nutrient Content of the Basal Diets (Experiment 2).

Items	1–21 Days	22–42 Days
Ingredient (%)		
Corn	57.0	61.9
Soybean meal (44.2%, crude protein)	31.3	25.6
Corn gluten meal (60%, crude protein)	3.9	4.3
Soybean oil	3.1	3.8
Dicalcium phosphate	1.8	1.6
Limestone	1.3	1.2
L-lysine	0.15	0.2
DL-methionine	0.15	0.1
Premix ^1^	1	1
Salt	0.3	0.3
Total	100	100
Calculation of nutrients		
Metabolizable energy, MJ/kg	12.69	13.10
Crude protein, %	21.52	19.71
lysine, %	1.14	1.04
methionine, %	0.50	0.43
Calcium, %	1.00	0.90
Available phosphorus, %	0.46	0.42
Arginine, %	1.36	1.19
Methionine+Cystine, %	0.85	0.76

^1^ provided per kg of diet: vitamin A (transretinyl acetate), 10,000 IU; vitamin D3 (cholecalciferol), 3000 IU; vitamin E (all-rac-α-tocopherol acetate), 30 IU; menadione, 1.3 mg; thiamin, 2.2 mg; riboflavin, 8 mg; nicotinamide, 40 mg; choline chloride, 600 mg; calcium pantothenate, 10 mg; pyridoxine•HCl, 4 mg; biotin, 0.04 mg; folic acid, 1 mg; vitamin B12 (cobalamin), 0.013 mg; Fe (from ferrous sulfate), 80 mg; Cu (from copper sulfate), 8 mg; Mn (from manganese sulfate), 110 mg; Zn (from zinc oxide), 65 mg; I (from calcium iodate), 1.1 mg; and Se (from sodium selenite), 0.3 mg.

**Table 2 animals-09-00953-t002:** Primers Used for Real-Time PCR (Experiment 2).

Genbank ID	Gene Name ^1^	Sequence (5′→3′) ^1^	Product Length
NM_205518	*β-Actin*	forward: TGCTGTGTTCCCATCTATCG	150
		reverse: TTGGTGACAATACCGTGTTCA	
NM_001031215.1	*CAT*	forward: GGTTCGGTGGGGTTGTCTTT	211
		reverse: CACCAGTGGTCAAGGCATCT	
NM_205064.1	*CuZnSOD*	forward: CCGGCTTGTCTGATGGAGAT	124
		reverse: TGCATCTTTTGGTCCACCGT	
NM_001277853.1	*GPx*	forward: GACCAACCCGCAGTACATCA	205
		reverse: GAGGTGCGGGCTTTCCTTTA	
HM237181.1	*HO-1*	forward: GGTCCCGAATGAATGCCCTTG	138
		reverse: ACCGTTCTCCTGGCTCTTGG	
NM_205117.1	*Nrf2*	forward: GATGTCACCCTGCCCTTAG	215
		reverse: CTGCCACCATGTTATTCC	

^1^*CAT*, catalase; *CuZnSOD*, copper and zinc superoxide dismutase; *GPx*, glutathione peroxidase; *HO-1*, heme oxygenase 1; *Nrf2*, NF-E2-related factor 2.

**Table 3 animals-09-00953-t003:** Effects of curcumin and bisdemethoxycurcumin on the growth performance in broilers (Experiment 2) ^1^.

Items		CON ^2^	CUR	BUR	*p*-Value
1-21 d	ADG, g/bird per day	34.85 ± 0.68	35.67 ± 0.64	35.41 ± 0.30	0.586
	ADFI, g/bird per day	49.78 ± 0.66	49.73 ± 0.93	48.89 ± 0.48	0.617
	FCR	1.43 ± 0.02	1.39 ± 0.02	1.38 ± 0.01	0.133
21-42 d	ADG, g/bird per day	89.93 ± 1.570 ^b^	96.79 ± 1.55 ^a^	93.35 ± 2.25 ^ab^	0.047
	ADFI, g/bird per day	164.31 ± 2.770	173.80 ± 2.88	167.81 ± 4.65	0.184
	FCR	1.83 ± 0.020	1.80 ± 0.03	1.80 ± 0.03	0.607
1-42 d	ADG, g/bird per day	59.42 ± 0.90 ^b^	63.08 ± 0.96 ^a^	61.31 ± 1.02 ^ab^	0.044
	ADFI, g/bird per day	100.09 ± 1.44	104.24 ± 1.68	100.95 ± 2.00	0.221
	FCR	1.69 ± 0.02	1.65 ± 0.02	1.65 ± 0.01	0.278

^1^ Values are expressed as the mean ± SEM (n = 8); Means with different letters (a-b) within the same row were significantly different (*p* < 0.05, by Test Tukey); ^2^ CON group: birds were fed the basal diet; CUR group: birds were fed the basal diet with 150 mg/kg curcumin; BUR group: birds were fed the basal diet with 150 mg/kg bisdemethoxycurcumin.

**Table 4 animals-09-00953-t004:** Effect of curcumin and bisdemethoxycurcumin on serum antioxidant capacity in broilers (Experiment 2) ^1^.

Items	CON ^2^	CUR	BUR	*p*-Value
CAT, U/mL	1.71 ± 0.24	1.95 ± 0.31	1.94 ± 0.45	0.854
T-SOD, U/mL	182.23 ± 12.05	202.78 ± 31.78	191.16 ± 12.88	0.788
GPx, U/mL	223.64 ± 12.71	231.71 ± 6.43	239.65 ± 16.59	0.674
T-AOC, U/mL	7.38 ± 0.67 ^b^	9.48 ± 0.45 ^ab^	10.15 ± 0.77 ^a^	0.016
T-SH, µmol/L	136.72 ± 11.13	139.94 ± 16.21	144.76 ± 10.25	0.905
MDA, nmol/mL	1.94 ± 0.08 ^a^	1.63 ± 0.08 ^b^	1.78 ± 0.05 ^ab^	0.018
PC, nmol/mg prot	1.11 ± 0.11	1.30 ± 0.17	1.14 ± 0.17	0.649

^1^ Values are expressed as the mean ± SEM (n = 8); Means with different letters (a-b) within the same row were significantly different (*p* < 0.05, by Test Tukey); ^2^ CON group: birds were fed the basal diet; CUR group: birds were fed the basal diet with 150 mg/kg curcumin; BUR group: birds were fed the basal diet with 150 mg/kg bisdemethoxycurcumin.

**Table 5 animals-09-00953-t005:** Effect of curcumin and bisdemethoxycurcumin on the redox potential of serum, liver and small intestine in broilers (Experiment 2) ^1^.

Items ^3^	CON ^2^	CUR	BUR	*p*-Value
Glutathione redox potential E_h_, mV				
serum	−81.91 ± 4.95	−90.57 ± 4.15	−87.49 ± 8.30	0.600
liver	−141.43 ± 1.05	−145.05 ± 1.61	−146.28 ± 1.45	0.057
duodenum	−121.46 ± 3.88	−122.79 ± 4.97	−127.23 ± 2.47	0.560
jejunum	−108.23 ± 3.57 ^a^	−121.79 ± 2.15 ^b^	−124.36 ± 0.81 ^b^	<0.001
ileum	−126.28 ± 1.40 ^a^	−131.02 ± 2.60 ^ab^	−134.86 ± 3.08 ^b^	0.060
r-GSH, μmol/L				
serum	1.04 ± 0.10	1.34 ± 0.12	1.33 ± 0.24	0.354
liver	124.41 ± 3.47 ^b^	135.46 ± 5.32 ^ab^	141.99 ± 4.96 ^a^	0.044
duodenum	77.44 ± 10.30	78.20 ± 10.36	83.60 ± 5.77	0.872
jejunum	58.14 ± 7.08 ^b^	80.39 ± 4.85 ^a^	91.47 ± 3.17 ^a^	0.001
ileum	103.20 ± 5.15	120.45 ± 8.27	131.47 ± 11.46	0.081
o-GSH, μmol/L				
serum	0.28 ± 0.05	0.23 ± 0.03	0.24 ± 0.05	0.672
liver	40.62 ± 1.60	36.50 ± 2.03	36.48 ± 1.74	0.196
duodenum	64.80 ± 2.68	57.69 ± 4.25	54.16 ± 4.47	0.167
jejunum	101.91 ± 3.06 ^a^	75.68 ± 4.97 ^b^	81.12 ± 2.92 ^b^	<0.001
ileum	88.04 ± 2.04	83.27 ± 5.49	72.97 ± 5.73	0.060

^1^ Values are expressed as the mean ± SEM (n = 8); Means with different letters (a-b) within the same row were significantly different (*p* < 0.05, by Test Tukey); ^2^ CON group: birds were fed the basal diet; CUR group: birds were fed the basal diet with 150 mg/kg curcumin; BUR group: birds were fed the basal diet with 150 mg/kg bisdemethoxycurcumin; ^3^ r-GSH, reduced glutathione; o-GSH, oxidized glutathione.

**Table 6 animals-09-00953-t006:** Effect of curcumin and bisdemethoxycurcumin on the antioxidant gene expression of liver, jejunum and ileum in broilers (Experiment 2) ^1^.

Items	CON ^2^	CUR	BUR
Liver, %			
*CAT* ^3^	100.00 ± 8.895	111.88 ± 8.681	113.85 ± 11.346
*HO-1*	100.00 ± 9.972	121.59 ± 6.569	129.41 ± 13.854
*GPx*	100.00 ± 9.930	112.21 ± 8.063	120.34 ± 7.935
*CuZnSOD*	100.00 ± 9.643	97.89 ± 4.701	110.61 ± 6.515
*Nrf2*	100.00 ± 4.032 ^b^	118.31 ± 7.744 ^ab^	121.32 ± 3.780 ^a^
jejunum, %			
*CAT*	100.00 ± 5.808	132.07 ± 8.004	144.11 ± 20.052
*HO-1*	100.00 ± 7.778 ^b^	139.24 ± 20.163 ^ab^	155.82 ± 15.376 ^a^
*GPx*	100.00 ± 12.405	127.60 ± 20.735	101.17 ± 7.485
*CuZnSOD*	100.00 ± 8.584	96.88 ± 6.349	124.40 ± 12.215
*Nrf2*	100.00 ± 7.663	105.65 ± 9.953	110.16 ± 8.740
ileum, %			
*CAT*	100.00 ± 8.136	97.93 ± 4.696	101.66 ± 4.998
*HO-1*	100.00 ± 6.306	99.76 ± 6.640	100.62 ± 9.090
*GPx*	100.00 ± 5.003	96.71 ± 3.639	98.62 ± 5.586
*CuZnSOD*	100.00 ± 8.776 ^b^	161.82 ± 11.127 ^a^	141.22 ± 9.167 ^a^
*Nrf2*	100.00 ± 5.907 ^b^	122.06 ± 6.876 ^a^	117.82 ± 5.374 ^ab^

^1^ Values are expressed as the mean ± SEM (n = 8); Means with different letters (a-b) within the same row were significantly different (*p* < 0.05, by Test Tukey); ^2^ CON group: birds were fed the basal diet; CUR group: birds were fed the basal diet with 150 mg/kg curcumin; BUR group: birds were fed the basal diet with 150 mg/kg bisdemethoxycurcumin; ^3^
*CAT*, catalase; *CuZnSOD*, copper and zinc superoxide dismutase; *GPx*, glutathione peroxidase; *HO-1*, heme oxygenase 1; *Nrf2*, NF-E2-related factor 2.

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
