# Peer review of "Comparative Studies on the Antioxidant Profiles of Curcumin and Bisdemethoxycurcumin in Erythrocytes and Broiler Chickens"

_animals, 2019, doi:10.3390/ani9110953_

Round 1

Reviewer 1 Report

MS ID: animlas-630564

The paper needs some modifications before publication.

The idea of this article is good, but there are some negative points such as…..

Simple summary:

Line 25: add ‘as well’ at the end of sentence (an animal production as well)

Introduction:

Line 75-76- and 92: in vitro and in vivo should be same pattern through the text, like italic in vitro

Line 90-92 : Please make two sentences for better understanding.  Start the sentence as follows

We further hypothesized that supplementation of curcumin …… (add that and of.. in due places)

MATERIAL AND METHODS

Line 176: Briefly, the serum was prepared as free from…… (add as in due place)

Line 192: liver sample , not only liver? ( add sample after liver)

Line 197: similar advises, add sample after : The expression of genes in liver, jejunum and ileum sample …

Line 197: replace was by were measured.

Results: Please use word (higher and lower) to compare the values, instant of using always  increase or decreased.

259-260: A trend towards an increased ADG (P > 0.05) from days 259 21 to 42 and days 1 to 42 was also observed in the BUR group?. Recheck the sig level? It should be same as like L 308-309.

Line 270: Use BUR supplementation

Line 273: among the tree treatments

Please use the p value in result table where suitable (Table 3,4,5)

Line 289: did not significantly differ ( add significantly not differ)

Table foot note p and text p is not same size and format, it should be same through the text and table.

Table 5: Mention the abbreviation of r-GSH and o-GSH in table footnote, , although it was explained in text but still needs to mention in table foot note (Table should be unique and independent).

L294-296: Delete: different; add among by replacing between.

L 308: were presented

L309-310: Rewrite the sentences: whereas an increase (P < 0.05) was only observed in the BUR group treatments for the liver Nrf2 mRNA level. ?

Discussion:

L349: in vivo

L352: was

L 353-354: which indicates that bisdemethoxycurcumin can be a potential candidate as naturalantioxidant to promote the animal health

L356-357: Sentence with Ref 24 can be delete from the text.

L383: in chickens

L403-404: delete the sentences

L411:Ref. number missing in text (Holder et al. reported, no Ref No. here?) Even this Ref is not found in List of Ref.

Include Ref number in text and list as well.

Conclusion

L422-424: the sentences related with discussion, delete it from conclusion.

L426: had the similar…….activities

Reviewer 2 Report

-minor changes are necessary:

Line 36: delete "the" in a total of 480 Arbor Acres broilers with THE similar body weights...

Line 67: In chemical systemS (add an s or add a "the" before chemical)

Line 228: marked hemolysis as compared TO the CON group (add a "to")

Line 325: were shown or are shown - correct english

Line 362: delete the unnecessary space before the literature reference

Line 369: is it referenced ot the results (then it should be they) or do you mean the study, then you should write the study also showed

Line 423: These antioxidant effects were associateD (add a d)

Author Response

This manuscript is a resubmission of an earlier submission. The following is a list of the peer review reports and author responses from that submission.

Round 1

Reviewer 1 Report

This paper deals with an interesting question and I believe it is relevant to the academic public and poultry sector. I do however have some comments and suggestions below, which I believe should be considered prior to re-submission. The description of the experiment is very detailed and well written. The English clear and understandable.

However, throughout the manuscript the Tables and Figures and not included. Thus, an evaluation of the manuscript, especially the results section, without this information is not possible.

Detailed Comments

TABLES AND FIGURES MISSING.

Introduction

Line 50 Studies, not studied Line 84-89: these are results stated in the introduction. You should delete this part.

Materials and Methods

Line 116: three independent experiments – do you mean replications? Line 142: which stunning method was used prior slaughter or killing by cervical dislocation? From an animal welfare point of view, it is necessary to stun the animals previously.

Results

Discussion

Line 267: we HAVE demonstrated (not has) Line 276: …due TO its role… (insert to) Line 285: … our results HAVE confirmed… Line 292: Curcuminoids…. HAVE been identified (not has)

Reviewer 2 Report

Comments to authors

MS ID: animals-580223

The paper needs some improvements before publication. At present the manuscript was prepared without result table and Figure in the main document.

The idea of this article is good, but there are some negative points such as…..

Simple summary:

Line 16: Scientific name should be italic.

Line 28: delete ‘that’ and ,

Line 28: insert ‘the’ before  bisdemethoxycurcumin as follows

 Our results demonstrated for the first time that, the bisdemethoxycurcumin, acting like ……….

Abstract:

Line 32: Mention the full abbreviation first (AAPH).

Introduction:

Line 48: Scientific name should be italic.

Line 49-50: the sentence should be rewritten.

Line 75: delete the word ‘animal”

Line 78: replace of  by on (on diet, excreta..)

Line 79: Use full meaning first then abbreviation, for ABTS and FRAP

Line 81-83: same advise for abbreviation,

Line 84-89: delete the results portion from this part, rather authors should state the hypothesis of both two experiments here.

MATERIAL AND METHODS

Line 92: delete the word-“kindly”

Line 110-111: use the author name as ref.

Line 117-118: Delete SOD and MDA from title (2.4) and use full abbreviation first in text and then Abbreviation in text.

Line 152: Feed conversion ratio(FCR)

Line 163-164: narrate the methodology briefly for some non-familiar techniques.

Line 169: Use appropriate synonym for reduced glutathione and oxidized glutathione

 GSH and GSSG ?? .

Line 175: liver sample , not only liver?

Line 180: recheck the language?

Line 194: Use Author name as Ref? in our previous study, could not be a well Ref.

Results:

Line 202: what does it means by AAPH, use full meaning first, in whole text while use first. Similar advices for other Symbol/ abbreviation.

Line 221: replace between by among

Result, Figure and table should be place in appropriate position, At present the manuscript was prepared without result table and Figure in the main document, I would like to revise the result portion (table and Figure) while author will resubmit the manuscript in revised format.

Discussion:

Line 262-264: Grammar error. Recheck the sentences

Line 264-266: Please rewrite the sentences

Line 268: improved

Line 268: enhanced

Line 272-274: authors should include here Ref to justify the findings.

Line 276-277: Please rewrite the sentences

Line 282-283: Please rewrite the sentences

Line 288-290: if the present findings are similar with the previous findings of the same authors, then question will arises why the authors did the same experiment? Please rewrite this portion.

Line 282-283: Please use the dose level?

Line 301-304: Please rewrite the sentences

Line 310: check the spelling ‘ of’

Line 323-324:rewrite the sentence

Line 338: Explain the abbreviation

Line 339-340:rewrite the sentence
